# A Novel Strategy for Screening Tumor-Specific Variable Domain of Heavy-Chain Antibodies

**DOI:** 10.3390/ijms241310804

**Published:** 2023-06-28

**Authors:** Abdur Rafique, Genki Hichiwa, Muhammad Feisal Jatnika, Yuji Ito

**Affiliations:** 1Graduate School of Science and Engineering, University of Kagoshima, Kagoshima 890-0065, Japan; k9939520@kadai.jp (A.R.); k9207801@kadai.jp (M.F.J.); 2Graduate School of Medical Sciences, Tottori University, Tottori 680-8550, Japan; d22m6009k@edu.tottori-u.ac.jp

**Keywords:** cancer, therapy, cell line, VHH, flow cytometry, next-generation sequencing, phylogenetic tree

## Abstract

The properties of the variable domain of heavy-chain (VHH) antibodies are particularly relevant in cancer therapy. To isolate tumor cell-specific VHH antibodies, VHH phage libraries were constructed from multiple tumor cells. After enriching the libraries against particular tumor cell lines, a next-generation sequencer was used to screen the pooled phages of each library for potential antibody candidates. Based on high amplification folds, 50 sequences from each library were used to construct phylogenetic trees. Several clusters with identical CDR3 were observed. Groups X, Y, and Z were assigned as common sequences among the different trees. These identical groups over the trees were considered to be cross-reactive antibodies. To obtain monoclonal antibodies, we assembled 200 sequences (top 50 sequences from each library) and rebuilt a combined molecular phylogenetic tree. Groups were categorized as A–G. For each group, we constructed a phagemid and determined its binding specificity with tumor cells. The phage-binding results were consistent with the phylogenetic tree-generated groups, which indicated particular tumor-specific clusters; identical groups showed cross-reactivity. The strategy used in the current study is effective for screening and isolating monoclonal antibodies. Specific antibodies can be identified, even when the target markers of cancer cells are unknown.

## 1. Introduction

Globally, tens of millions of people are diagnosed with various types of cancer each year. Cancer is the second leading cause of death worldwide, accounting for approximately 10 million deaths every year. The cancer burden can be reduced through early detection and effective treatment [1,2,3]. Various molecular biology methods have been developed for cancer detection and treatment, including molecular-targeted therapies. Molecular-targeted therapies have a potential role in enhancing the long-term survival of patients with cancer [4,5,6]. However, cancer-cell-specific antibodies are required to develop targeted biologics.

Antibody discovery remains the biggest challenge because selected antibodies are required to recognize cancer cell molecular targets, such as signaling receptors [7,8]. For the diagnosis and treatment of cancers, antibodies engineered from the variable domain of heavy-chain (VHH) antibody have gained significant attention because it contains several outstanding properties that favor novel biologics. VHH antibodies are attractive candidates due to their small size (which allows for improved penetration), unique ability to independently bind foreign targets, heat (>80 °C) resistance, and chemical resistance [9,10,11,12].

Antibody phage display is an in vitro antibody selection technology successful in isolating affinity peptides and antibodies with desired functions from diverse gene libraries. During phage display, phage particles are trapped by in vitro enrichment methods, bearing billions of foreign fusion peptides or antibodies, and interact with immobilized targets in a stationary phase under controlled conditions [13,14,15]. Various in vitro platforms are available to enrich the phage library, including immuno tubes, plastic plates, and magnetic-bead panning. However, the use of the platforms is limited due to improper folding of antigens during absorption on their surfaces. In contrast, whole-cell panning, using a cell cytometer, is one of the best strategies for successfully isolating antibodies because it allows the representation of membrane proteins in their native conformation [16].

The major challenge is isolating target-specific VHH antibodies through conventional screening of phage libraries, using phage ELISA or phage-binding analysis. Conventional screening requires the use of thousands of library clones and generates a monoclonal phage for binding analysis, which is very laborious, time-consuming, and expensive [17]. In addition, hetero-immunized library genes could be stimulated by multiple responses, due to shared antigenic determinants, with multiple VHH antibodies [18]. To isolate mono-specific antibodies, comparative binding analysis is necessary. Therefore, we searched for an alternative antibody screening strategy and found that a high-throughput sequencing-based strategy is an ideal approach for screening antibody candidates from a diverse gene library using machine-learning informatics. High-throughput sequencing via next-generation sequencing (NGS) is a powerful tool for analyzing a large number of DNA sequences with many evolutionary aspects of phage libraries within a short time frame, including the identification of unique and/or low-frequency rare clones that could be missed by conventional isolation techniques [14,19,20]. In principle, NGS-generated reads are used to determine the enrichment folds of each sequence by comparing them before and after panning of the phage library. High amplification folds indicated that the number of identified reads had increased compared to the number in the previous selection rounds. Diversified NGS-generated sequences can be grouped according to evolutionary relationships by a phylogenetic tree that provides a highly homologous cluster of genes derived from one or a group of species [21,22]. The phylogenetic tree-generated cluster would contain homologous character populations that share a recent common ancestor. Based on this principle, we searched monoclonal VHH antibodies from multiple tumor cells immunized phage display library.

In the current study, we aimed to develop a tumor-specific VHH antibody screening method to screen multiple tumor cell-immunized libraries, using high-throughput sequencing, which will help us to identify antibody candidates.

## 2. Results

### 2.1. Construction of Phage Libraries and Enrichment of Phages through Cell Panning

Phage libraries from short (IgG2-derived) and long hinge (IgG3-derived) VHH were independently constructed with an estimated size of 1.9 × 10^7^ cfu/500 ng of IgG2-derived VHH DNA inserted phagemid and 4.6 × 10^7^ cfu/500 ng of IgG3-derived VHH DNA inserted phagemid, respectively. Phage libraries were enriched through whole-cell panning using a cell cytometer (Appendix A). To distinguish between target and absorptive cells during sorting, target cells were stained with calcein-AM to prepare fluorescent shifts, and three other cells were mixed with the phage library for non-specific phage absorption (Figure 1 and Table 1). The enrichment operations were performed in duplicate for S1T, HepG2, and SKBR-3 libraries; and triplicate for the KYSE520 library. The panning rounds of phage binding for each library were analyzed using FACS (fluorescence is proportional to phage enrichment) (Figure 2). Phage enrichment was confirmed by a substantial increase in fluorescence intensity after panning against the tumor cell lines. In the S1T library, the enrichment of each panning round was successively shifted; however, in other cell lines, the enrichment was unchanged after the first round of panning.

### 2.2. High-Throughput Sequencing Analysis

To identify tumor-specific VHH antibodies, we performed high-throughput sequencing analysis of phage libraries before and after cell panning (summarized in Table 2). The frequencies (%) of the VHH genes in the phage libraries before and after panning were compared by NGS, and the changes in the frequencies of every VHH sequence through panning were calculated. The change in frequency, defined as the amplification factor (fold), reflects the efficiency of increasing the number of individual VHH clones through panning. The highest estimated amplification folds for each library were as follows: S1T, 4607.79; SKBR-3, 13,447.25; HepG2, 1820.39; and KYSE520, 7267.25 (Appendix A). Based on the amplification folds, we aligned the top 50 VHH sequences from each library and constructed individual molecular phylogenetic trees, using the neighbor-joining method (Figure 3 and Appendix A). Numerous clusters appeared in each library, with varying numbers of populations in a homologous cluster. During the CDR3 comparison of 200 sequences (top 50 sequences from each library), identical groups of CRD3s were observed in more than one library with high amplification folds, named groups X, Y, and Z (Table 3 and Appendix A). We changed the strategy for screening monoclonal VHH antibody binders and constructed a combined molecular phylogenetic tree using all 200 sequences that were categorized into six new clusters with a high number of homologous populations, named groups A to G, including previously observed identical groups X, Y, and Z (Figure 4). Groups A to G appear to be mono-specific binders, whereas groups X, Y, and Z appear to be multi-specific binders, according to the homogeneity of cluster populations.

### 2.3. Binding Properties of NGS Identified Groups of VHH Clones

Groups of VHH clone phagemids were reconstructed by overlapping PCR, using CDR3-specific primers as shown in Appendix A. Phages were produced for the representative clones from groups A (HepG2_1), B (S1T_3), C (SKBR3_13), D (SKBR3_6), E (KYSE520_2), F (SKBR3_2), and G (KYSE520_1). Phage-binding ability to S1T, SKBR3, HepG2, and KYSE520 cell lines was determined using FACS. Groups A to F showed highly specific binding toward HepG2, S1T, SKBR3, SKBR3, KYSE520, and SKBR3 tumor cells, respectively. The Group G phage displayed multi-specific tumor cell binders, even though it was obtained from KYSE520 specific homologous cluster (Figure 5).

Similarly, identical groups of X (S1T_14), Y (S1T_2), and Z (S1T_ 4) clone (Table 3) phages were constructed and their binding specificity to four tumor cells was determined. The identical CDR3 generated Group X displayed binding to all cell lines, Group Y to S1T and HepG2, and Group Z to S1T, SKBR-3, and KYSE520 (Figure 6).

## 3. Discussion

Many therapeutic antibodies are available for treating cancers with various indications, including breast, colorectal, lung, oral, and oropharyngeal cancers. However, the use of these antibodies is limited due to their large size, poor tissue penetration, non-specific binding, low affinity, and poor stability, which may cause side effects on the human body [23,24].

In our laboratory, we are currently focusing on the development of VHH antibodies, considered next-generation cancer therapeutics. In the current study, we aimed to improve the effectiveness of antibody therapeutics and detection accuracy by improving the isolation and screening strategies for hetero-immunized alpaca VHH antibody libraries. We propagated library phages and subjected them to whole-cell panning against S1T, SKBR-3, KYSE520, and HepG2 cell lines to construct tumor-specific enriched phage libraries, using flow cytometry. Enriching library phages against target molecules on specific tumor cells can be challenging due to the abundance of non-target organic molecules on the cell surface [25]. To minimize non-target phages, we applied the subtraction method in cell panning procedures, which involved the use of three cell lines (other than the target cell line) to absorb non-target-specific phages. In addition, target cells were stained with calcein-AM to prepare a fluorescence shift and distinction from absorbent cells [26,27]. Based on this principle, we enriched phage libraries and confirmed the enrichment of each library by phage binding to the target cell line (S1T, SKBR3, KYSE520, and HepG2).

Hetero-immunized library-generated antibodies have a greater chance of cross-reactivity because antigenic determinants are shared by multiple antibodies during immunization through multiple antigens or tumor cells [18], where conventional screening of monoclonal antibodies will be faced with difficulties that will lead to enhanced cost, labor, and time. To overcome the challenges associated with conventional screening, we comprehensively analyzed the pooled phage libraries using a next-generation sequencer [28,29]. The NGS-generated reads were assembled, and the amplification folds for each library were determined using Sequence Ordering Program Ranked by Amplification (SOPRA), where sequences were ranked based on their corresponding amplification factors [14]. High amplification factors indicate a high frequency in the libraries after the enrichment process. The highest amplification fold of each library was 4601.79 (S1T), 13,447.25 (SKBR-3), 1820.39 (HepG2), and 7267.25 (KYSE520). We aligned the top 50 sequences for each library according to descending amplification folds, which are highly applicable for high-throughput screening using phylogenetic trees [30]. The tumor-specific phylogenetic trees were categorized into several typical clusters with varying numbers of populations and a low degree of homology, according to genetic distances [31].

Therefore, to compare the homology of all libraries CDR3, we combined 200 sequences (top 50 sequences from each library) and aligned several groups of identical CDR3 with 98% sequence homology derived from more than one library read, originating in groups X, Y, and Z. As CDR3 is the dominant antigen binding contributor [32], the identical CDR3 containing VHH sequences showed high amplification folds: S1T_14 from Group X, 446; S1T_2 from groups Y, 4254; and S1T_4 from Group Z, 3277. These findings suggest that cross-reactive binders are retained in high-amplification folds.

To overcome this issue, the classification of cross-reactive and mono-specific VHH antibody clones, according to evaluation distances using a molecular phylogenetic tree, was necessary. To filter the cross-binders, we constructed a combined 200-sequence phylogenetic tree where several clusters were generated and categorized into groups A to G with 98.8% sequence homology with cross-reactive clusters, groups X, Y, and Z.

The representative VHH clones (HepG2_1) in Group A showed an amplification fold factor of 1820.39. The VHH (S1T_3, SKBR3_13, SKBR3_6, KYSE520_2, SKBR3_2, and KYSE520_1) amplification folds from the groups B, C, D, E, F, and G were 4093.19, 567.50, 978, 6046.70, 10,077.37, and 7267.25, respectively. These were used to generate monoclonal phages and we determined the monoclonal phage binding towards four tumor cell lines. Group A to F phage binding indicated monoclonal or tumor-specific antibodies, which confirmed the validation of high-throughput screening followed by phylogenetic tree analysis. The cross-reactivity of the group G phage was confirmed by demonstrating its binding with multiple tumor cells. The most probable reason for the cross-reactivity of the Group G cluster is that it contained the sub-cluster population of HepG2 and the S1T library, which means they were derived from the nearest common ancestor. Another possible reason is that an insufficient number of NGS-generated sequences were considered for the phylogenetic tree from each library during cluster generation.

To investigate identical groups of VHH clones, randomly reconstructed S1T_14 from X, S1T_2 from Y, and S1T_6 from Z clone phagemids and propagated phages were used. Group X and Z phages displayed binding ability with four and three tumor cells, although they belonged to triple and double cell lines, respectively. The phage binding of Group Y displayed cross-reactive binding with S1T and HepG2 cell lines with a frequency of 2.533% (4253,91 amplification folds) and 0.0483% (62.73 amplification folds) with the S1T and the HepG2 libraries, respectively.

In addition to phage binding towards tumor-specific cells, HepG2 cells showed partial positive binding. These partial bindings could be considered insignificant because many studies have reported that partial binding between HepG2 and heterologous monoclonal antibodies often occurs, as the HepG2 cell line does not have specific cancer markers and shares some biological surface proteins expressed in most normal epithelial tissues, lymphatic tissues, and cancer stem cells, including cells of HNSCC, breast, colon, liver, ovarian, pancreatic, and gastric cancer [33,34,35,36].

This study had some limitations. Due to the lack of previous studies on this topic, we could not optimally design antibodies to promote the identification of target antigens for each isolated antibody whether mono-specific or multi-specific. Moreover, due to limited NGS skills, we may have missed target antibody genes or superior therapeutic candidates.

Nevertheless, it can be concluded that the isolation strategy, using high-throughput sequencing and screening, for the discovery of mono-specific and multi-specific VHH antibodies is highly efficient and crucial for the preparation of next-generation antibody therapeutics as well as research and diagnostics.

## 4. Materials and Methods

### 4.1. Alpaca Immunization

All animal immunization experiments were approved by the ethics committee of ARK Resource (Kumamoto) under approval number AW-16039. The experiments were performed at ARK Resource (Kumamoto), according to a defined protocol (161-031) on July 19, 2016. Briefly, 1.0 × 10^7^ cells of each cancer cell type (SK-BR3, Hep-G2, KYSE-520, and S1T) were suspended in 0.5 mL phosphate-buffered saline (PBS). The cells were inactivated by heat treatment at 60 °C for 10 min complete Freud’s adjuvant was used for the first immunization, Freund’s, and Freund’s incomplete adjuvant were used for immunizations 2 to 4. Immunization was performed at an interval of 2 weeks. After immunization, 50 mL of blood was collected. Peripheral blood lymphocytes were isolated and homogenized using RNAiso Plus (Takara Bio Inc., Shiga, Japan, according to the manufacturer’s protocol, and stored at −80 °C.

### 4.2. Construction of Tumor Cell-Specific VHH Phage Library

Total RNA was extracted from the alpaca PBMC homogenate using RNAiso Plus (Takara Bio), according to the manufacturer’s protocol. cDNA was synthesized from 4.5 µg total RNA by reverse transcriptase using the Oligo (dT)20 primer and the SuperScriptTM III First-Strand Synthesis System for RT-PCR (Invitrogen, Carlsbad, CA, USA). VHH gene amplification was carried out using 100 ng of cDNA as a template, with the common forward VHH-specific 5′-AGKTGCAGCTCGTGGAGTCNGGNGG-3′and reverse (IgG2) short hinge-specific primers, 5′-GGGGTCTTCGCTGTGGTGCG-3′ or (IgG3) long hinge-specific primer 5′-TTGTGGTTTTGGTGTCTTGGG-3′. The first PCR was performed in a 50 µL reaction mixture with Gene Tag DNA polymerase (Nippon Gene Co., Ltd., Toyama, Japan). The reaction steps were an initial denaturation step (98 °C for 2 min), followed by 22 repetitions of a three-step cycle: denaturation (98 °C for 30 s), annealing (58 °C for 30 s), and extension (72 °C for 1 min).

For the 2nd PCR, 10 ng of the 1st PCR product was used as the template. The PCR was performed to add the restriction sites at both ends of the VHH gene. The common forward primer harboring the Sfi I site (5′-TGCTCCTCGCGGCCCAGCCGGCCATGGCTCAGGTGCAGCTCGTGGAGTCTGG-3′) and the reverse (IgG2) short hinge-specific primer (5′-ATGATGATGTGCACTAGTGGGGTCTTCGCTGTGGTGCG-3′) or the (IgG3) long hinge-specific primer (5′-ATGATGATGTGCACTAGTTTGTGGTTTTGGTGTCTTGGG-3′) both harboring the Spe I site were used. The second PCR was performed using Gene Taq DNA polymerase (Nippon Gene Co., Ltd.). The reaction steps were an initial denaturation (98 °C for 2 min), followed by 25 repetitions of a three-step cycle: denaturation (98 °C for 30 s), annealing (58 °C for 30 s), and extension (72 °C for 1 min).

VHH libraries were constructed by digestion of the VHH gene and phagemid vector (pKSTV-03), using SfiI and SpeI restriction enzymes (New England Biolabs, Tokyo, Japan). The digestion products were recovered through an agarose gel cut process, and the vector and VHH insert were ligated using T4 DNA ligase (Nippon Gene Co. Ltd.). The constructed phagemid was purified by phenol-chloroform treatment, and 500 ng phagemid was electroporated into *Escherichia. Coli* TG1 electrocompetent cells using a MicroPulser (Bio-Rad, Hercules, CA, USA). A portion of electroporated TG1 was used to estimate library diversity, and the remainder was spread on a 2TYAG plate to make a library stock.

### 4.3. Phage Propagation

Five hundred microliters of the thawed library glycerol stock (TG1 cells) were added to 500 mL of 2TY (Tryptone-Yeast) medium with 50 mg ampicillin and 10 g of glucose. The cells were incubated at 37 °C with shaking at 220 rpm until the OD600 reached 0.5–0.6. Superinfection was performed using the M13KO7 helper phage (Invitrogen) at a multiplicity of infection (MOI) of 20 and incubated at 37 °C for 30 min, followed by 30 min at 37 °C with shaking. The bacterial cells were centrifuged at 1500× *g* for 10 min, and the supernatant containing excess uninfected helper phage was discarded. The *E. coli* pellets infected with the helper phage were inoculated into fresh 2 TYAK (A: ampicillin and K: kanamycin, FUJIFILM Wako Pure Chemical Corporation, Osaka, Japan) medium and cultured at 37 °C for 16 h. The culture solution was centrifuged at 7000× *g* for 20 min and the phage solution suspended in the supernatant was recovered with the addition of 0.2 vol PEG/NaCl, which results in a phage precipitation reaction. Finally, the solution was centrifuged at 12,000× *g* for 60 min, and the supernatant was discarded (any residual supernatant was completely removed using a pipette). The phage pellets were resuspended in 4 mL of PBS, filtered, and stored at 4 °C until further use.

### 4.4. Phage Library Enrichment by Cell Sorter

S1T is a leukemia cell line isolated from patients with adult T-cell leukemia-lymphoma (ATL), SKBR3 is a human breast cancer cell line isolated from female adenocarcinoma patients, KYSE520 is a human esophageal squamous carcinoma cell line isolated from the lower intra-thoracic esophagus, and HepG2 is a pure cell line of human liver carcinoma. All cell lines were purchased from ATCC (Manassas, VA, USA). A commercially available culture medium was used to grow all cell lines at 37 °C and 5% CO_2_. The S1T cell line was cultured in RPMI-1640 supplemented with 10% FBS and 1% streptomycin-penicillin. SKBR3 was cultured in McCoy’s 5A. KYSE520 and HepG2 cell lines were cultured in D-MEM. A total of 6.0 × 10^6^ cells/1 mL, (2.0 × 10^6^ cells for each cell line other than target cells) were centrifuged at 200× *g* for 3 min at 4 °C. The supernatant was removed and the pellet was washed with 2% BSA/PBS, and completely resuspended in 1 mL of pfu library phage (1.0 × 10^11^). The cell-phage solution was maintained at 4 °C for 30 min to allow for non-target binding.

Target cells 2.0 × 10^6^ cells/1 mL PBS were dispersed in 1 mL of 100 nM calcein-AM solution (DOJINDO), and fluorescently labeled reactions were performed at 37 °C for 15 min. The cells were washed twice with 1 mL of 2% BSA/PBS. The absorption cells (pre-reacted with library phages) were mixed with the fluorescently labeled target cells and incubated at 4 °C for 1 h to allow phage binding to target cells. To remove unbound phage, the cells were washed twice with 1 mL of 2% BSA/PBS and then suspended in 1 mL PBS.

For dead cell staining, 5 µL of 7-AAD Viability Dye (Bio-Rad) was added to the cell-phage solution. Then, the cells were incubated at 4 °C for 15 min, washed with BSA/PBS, and suspended in 2 mL of PBS. Subsequently, the cells were mesh-filtered and injected into the cell sorter S3e (Bio-Rad), which automatically sorted and collected 7-AAD-negative and calcein-positive cells.

The collected cells were centrifuged at 4 °C at 300× *g* for 3 min and the supernatant was removed. The bound phage was dissociated from the cells by the addition of 0.1 M Glycine-HCI (pH 2.2). After 5 min, the solution was neutralized with IM Tris-HCI (pH 9.0). The rescued phages were infected with *E. coli* TG-1 (Lucigen, Middleton, Wisconsin, USA) and cultured on 2TYAG plates at 30 °C for 12 h, to generate library stock. A portion of the infected *E. coli* was used to measure the enriched library size. Target cell-specific phage library enrichment was performed by repeating this procedure.

### 4.5. Enriched Phage Binding with Tumor-Specific Cells by FACS

The binding activity of the polyclonal phage libraries was determined after whole-cell panning, using a flow cytometer. 2.0 × 10^5^ cells/100 μL of each cell line was dispersed in 2% BSA/PBS and mixed with 100 μL of 1.0 × 10^11^ pfu phage in each panning round. The reaction mixture was incubated at 4 °C for 1 h. The cells were then washed three times with 2% BSA/PBS, resuspended in 100 µL solution mixture of 100 ng mouse anti-M13- mAb-biotin (Abcam, Cambridge, UK), pre-labeled with 200 ng streptavidin-PE (Miltenyi Biotech, Bergisch Gladbach, Germany), incubated at 4 °C for 30 min, and then washed three times with 2% BSA PBS. Finally, the cell pellets were resuspended in 500 µL of PBS, and the binding ability was determined using a flow cytometer (Bio-Rad).

Monoclonal phage binding was performed after reconstructed NGS identified a representative phage clone. A total of 2.0 × 10^5^ cells/100 μL of each cell line was dispersed in 2% BSA/PBS and mixed with 100 μL of 1.0 × 10^11^ pfu phage in each representative clone phage. The reaction mixture was incubated at 4 °C for 1 h. The cells were then washed three times with 2% BSA/PBS. The cells were resuspended in a 100 µL solution mixture of 100 ng of mouse anti-M13- mAb-biotin (Abcam) pre-labeled with 200 ng streptavidin-PE (Miltenyi Biotech, Germany), incubated at 4 °C for 30 min, and then washed three times with 2% BSA/PBS. Finally, the cell pellets were resuspended in 500 µL of PBS, and the binding ability was determined using a flow cytometer (Bio-Rad).

### 4.6. NGS Sample Preparation and Analysis

VHH gene sequences, in the phage library, were analyzed using a MiSeq system (Illumina, Inc., San Diego, CA, USA). The MiSeq library for DNA sequencing was prepared using a QIAseq 1-step Amplicon Library Kit (QIAGEN), according to the manufacturer’s protocol. Phagemids were extracted from 0-round, 2nd round of S1T, SKBR3, and HepG2, and 3rd round of KYSE520. The DNA sequencing samples were PCR amplified using primers for both Alpaca VHH IgG2 and IgG3, the common forward primer 5′-GGTGCAGCTCGTGGAGTCTGGGGG-3′ and reverse primers for IgG25 5′-GGGGTCTTCGCTGTGGTGCGC-3′ and IgG3 5′-GTGGTTTTGGTGTCTTGGGTTC-3′. The final loading concentration was adjusted to 15 pM using the MiSeq loading protocol. The MiSeq Reagent Kit v3 was used for long paired-end (2 × 300 bp) sequencing reactions. The run quality was monitored following the standard Illumina procedure. The error rate was estimated using a control DNA sequence that was sequenced parallel to the samples. The sequencing reads were assigned to a raw data pool, based on a unique 8-bp barcode identifier, and were generated as a FASTQ file. The raw paired-end nucleotide sequences were merged, filtered, aligned, and trimmed using USEARCH Ver.8.0 (Edgar, RC) to remove low-quality and meaningless short sequences. Subsequently, an analysis program called the SOPRA, which is described by a custom Perl script was used to align, cluster, and count amino acid sequence data in the VHH phage display library. The phylogenetic tree was constructed from VHH amino acid sequence data using the neighbor-joining method and CLC Genomics Workbench 9 (QIAGEN Bioinformatics, Maryland, USA).

### 4.7. Regeneration of NGS-Identified Clones and Binding Analysis

We designed a CDR3-specific overlapping PCR primer for each group (Appendix A) and regenerated the NGS-identified groups of the VHH gene from the pooled phage library using overlapping PCR techniques. The phagemids were constructed by enzymatic digestion and ligation. The ligation products were transfected into *Escherichia coli* (TG1) (Lucigen, Middleton, Wisconsin, USA.). Using transfected TG1, the monoclonal phage was propagated and the phage-binding ability towards tumor-specific cells was determined using FACS.

## Figures and Tables

**Figure 1 ijms-24-10804-f001:**
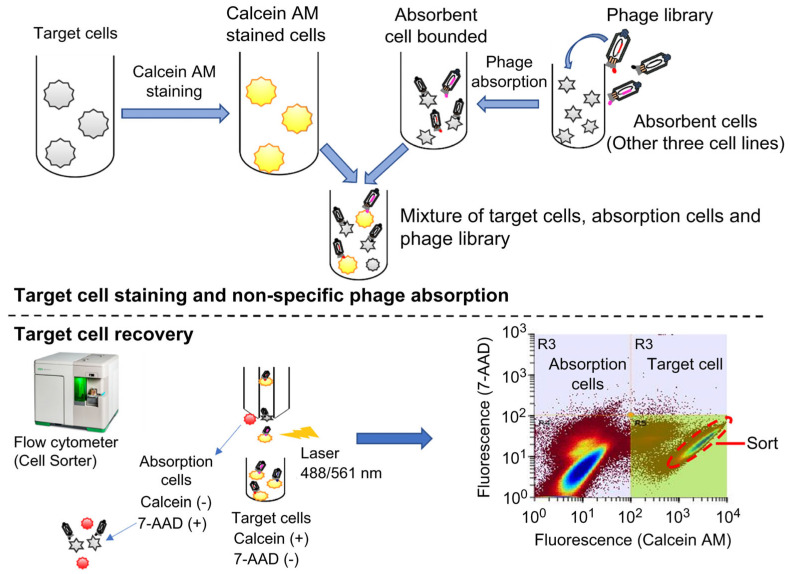
Overview of phage library enrichment through whole-cell panning. Preparation of fluorescently labeled target cells stained with calcein-AM and absorption cells for non-specific phage binding (**upper panel**). A mixture of target and absorption cells was observed using a cell cytometer to sort the phage-binding-shifted target cells (**lower panel**). In gated channels, absorption cells exhibited a high population density (**left**) whereas target cells exhibited a low population density (**right**).

**Figure 2 ijms-24-10804-f002:**
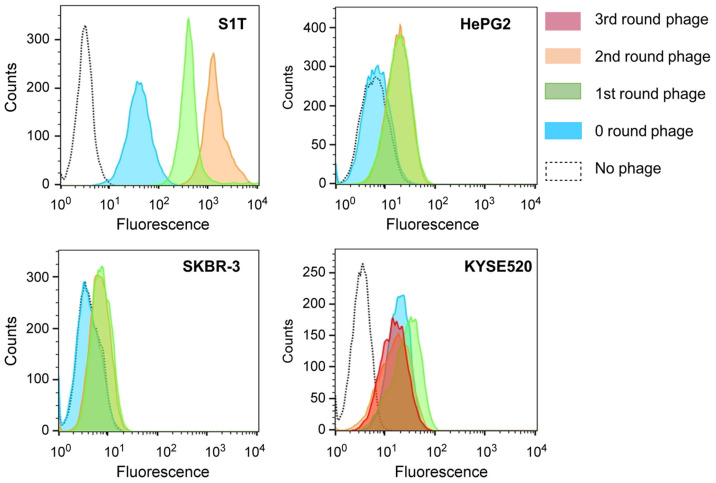
Polyclonal phage binding for library enrichment. Phage libraries were enriched for S1T, HepG2, SK-BR3, and KYSE 520 cell lines. Phage library enrichment was repeated twice for S1T, HepG2, and SK-BR3; and thrice for KYSE 520 cell lines. Colored boxes indicate the enrichment rounds of the phage library. The *X*-axis and *Y*-axis indicate the fluorescence intensity and the number of cells, respectively.

**Figure 3 ijms-24-10804-f003:**
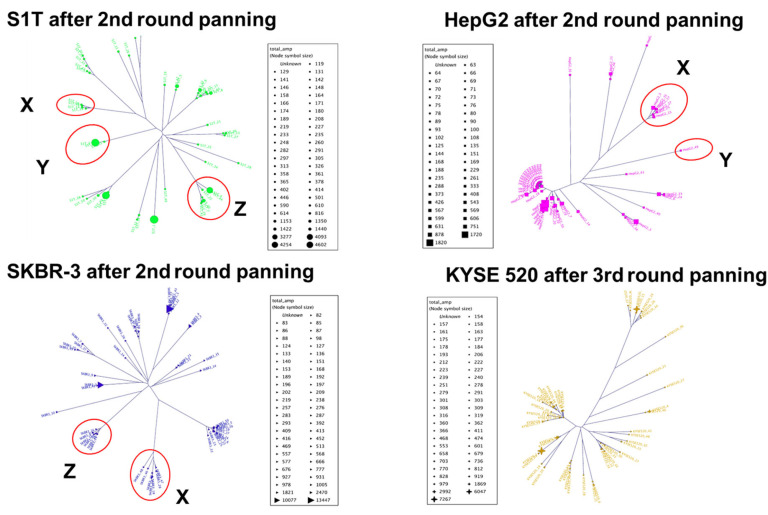
Construction of tumor-cell specific library phylogenetic tree. Based on high amplification folds, we selected 50 sequences from each library to construct a phylogenetic tree specific to the tumor cell line. The red circles indicate groups X, Y, and X denoting identical sequences.

**Figure 4 ijms-24-10804-f004:**
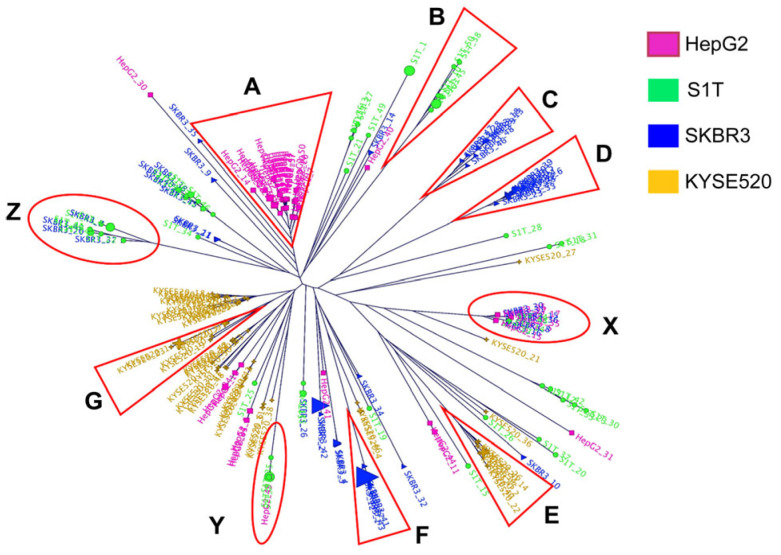
Combined library molecular phylogenetic tree. Two hundred sequences (top 50 sequences from each library) were combined and a molecular phylogenetic tree was constructed using the neighbor-joining method. Red triangles indicate the highly homologous groups A to G and red circles indicate the clustered heterologous groups X, Y, and Z. The groups of sequences were identified by an independent tumor cell library phylogenetic tree. The pink, green, blue, and golden colors represent the HepG2, S1T, SKBR-3, and KYSE520 libraries, respectively. These NGS data sequences were deposited in the DDBJ Sequence Read Archive (DRA) under accession number DRA014254.

**Figure 5 ijms-24-10804-f005:**
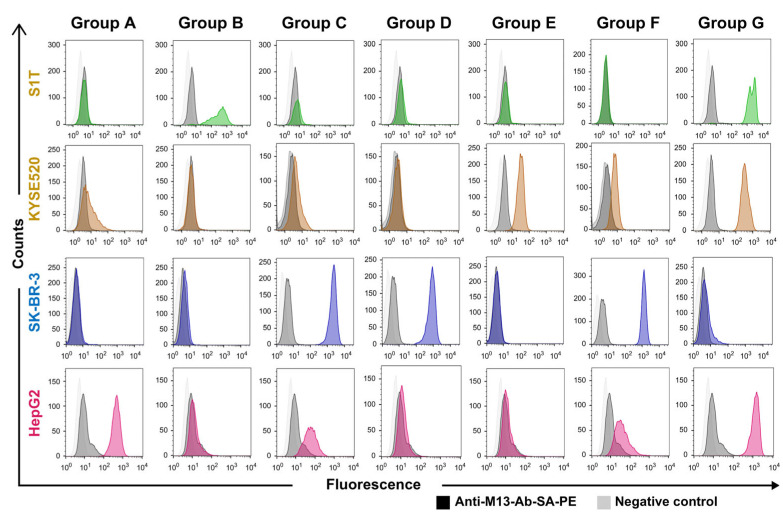
Combined phylogenetic tree-generated groups of representative clone phage binding to tumor cells. The *Y*-axis indicates the number of detected cells, and the *X*-axis indicates the fluorescence intensity. Each column of this diagram refers to groups of phages and rows refer to tumor cell lines. Colors represent the cell lines; green: S1T cell line, brown: KYSE520 cell line, blue: SKBR-3 cell line, and pink: HepG2 cell line. Light and deep gray colors represent untreated cancer cell lines and secondary antibody-treated cell lines, respectively.

**Figure 6 ijms-24-10804-f006:**
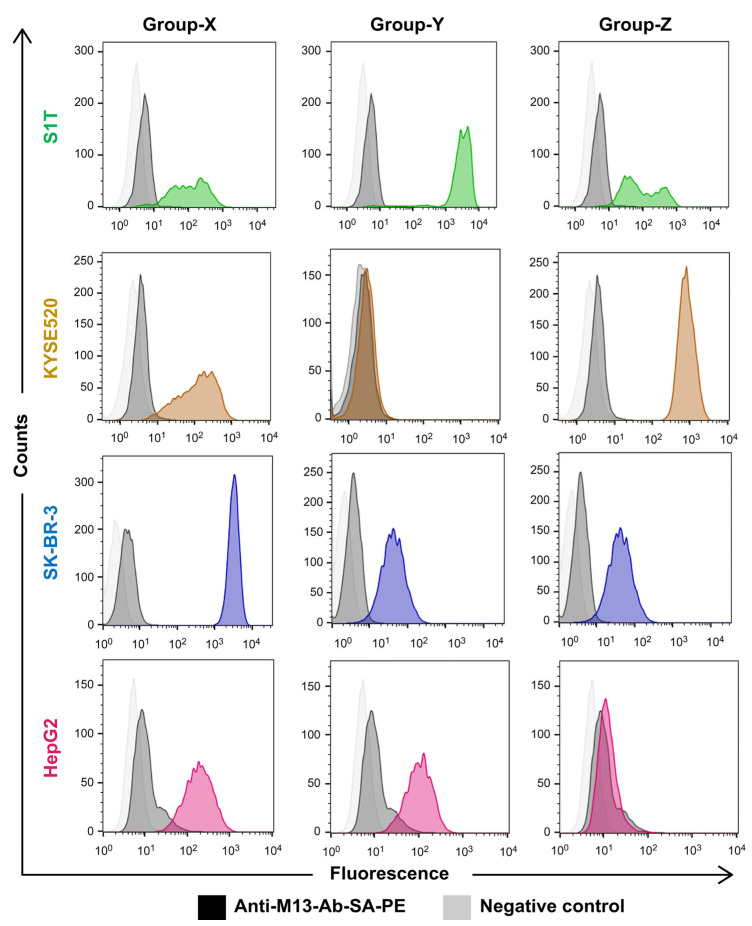
Tumor-cell specific phylogenetic tree-generated identical groups of representative clone phage binding to tumor cells. The *Y*-axis represents the number of detected cells, whereas the *X*-axis indicates the fluorescence intensity. Each column in the diagram represents a group of phages, and the rows correspond to tumor cell lines. The color scheme is as follows: green for S1T cell line, brown for the KYSE520 cell line, blue for the SKBR-3 cell line, and pink for the HepG2 cell line. Light and deep gray colors represent untreated cancer cell lines and secondary antibody-treated cell lines, respectively.

**Table 1 ijms-24-10804-t001:** Non-specific phage absorbent cell lines for each individual library during cell panning.

Target Cell Line	1st Round Absorbent Cell Line	2nd Round Absorbent Cell Line	3rd Round Absorbent Cell Line
S1T 2.0 × 10^6^ cell	HepG2, SKBR-3, KYSE520 6.0 × 10^6^ cell (2.0 × 10^6^ cell/cell line)	HepG2, SKBR-3, KYSE520 6.0 × 10^6^ cell (2.0 × 10^6^ cell/cell line)	
KYSE520 2.0 × 10^6^ cell	HepG2, SKBR-3, S1T 6.0 × 10^6^ cell (2.0 × 10^6^ cell/cell line)	HepG2, SKBR-3, S1T 6.0 × 10^6^ cell (2.0 × 10^6^ cell/cell line)	HepG2, SKBR-3, S1T 6.0 × 10^6^ cell (2.0 × 10^6^ cell/cell line)
SKBR-3 2.0 × 10^6^ cell	HepG2, S1T, KYSE520 6.0 × 10^6^ cell (2.0 × 10^6^ cell/cell line)	HepG2, S1T, KYSE520 6.0 × 10^6^ cell (2.0 × 10^6^ cell/cell line)	
HepG2 2.0 × 10^6^ cell	S1T, SKBR-3, KYSE520 6.0 × 10^6^ cell (2.0 × 10^6^ cell/cell line)	S1T, SKBR-3, KYSE520 6.0 × 10^6^ cell (2.0 × 10^6^ cell/cell line)	

**Table 2 ijms-24-10804-t002:** Summary of high-throughput sequencing results.

Parameters	‘0′ Round	2nd Round S1T Library	2nd Round HepG2 Library	2nd Round SKBR-3 Library	3rd Round KYSE 520 Library
Total number of read sequences	398,055	471,933	354,002	499,052	261,839
Number of merged sequences	116,952 (100%)	186,946 (100%)	129,050 (100%)	209,908 (100%)	94,059 (100%)
Single occurrence sequences	31,497 (26.93%)	26,685 (14.27%)	4837 (3.75%)	9397 (4.48%)	10,082 (10.72%)
Unique sequence	37,206 (31.81%)	32,652 (17.47%)	6958 (5.39%)	13,733 (6.54%)	13,358 (14.20%)
Highest frequency	17,611 (15.06%)	22,471 (12.02%)	61,377 (47.56%)	42,054 (20.03%)	6940 (7.38%)

These NGS data sequences were deposited in the DDBJ Sequence Read Archive (DRA) under accession number DRA014254.

**Table 3 ijms-24-10804-t003:** Identical CDR3s clones identified by high-throughput sequencing.

Identical CDR3 Sequences (Group)	S1T Library Clone Number (S1T)	HepG2 Library Clone Number (HepG2)	SKBR3 Library Clone Number (SKBR3)	KYSE 520 Library Clone Number (KYSE520)
Group X	14, 24 & 39	5, 15, 17, 25, 27, & 35	8, 15, 16, 30, 31, & 36	
Group Y	2, 9 & 35	49		
Group Z	4, 7, 22, 29, 40, 41, & 44		7, 20, 37, & 49	

## Data Availability

The NGS data sequences were deposited in the DDBJ Sequence Read Archive (DRA) under the accession number DRA014254.

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
