# Peer review of "A Novel Strategy for Screening Tumor-Specific Variable Domain of Heavy-Chain Antibodies"

_ijms, 2023, doi:10.3390/ijms241310804_

Round 1

Reviewer 1 Report

The manuscript “A novel strategy for screening tumor-specific variable domain of heavy-chain antibodies” by Abdur et al. is a well-designed and presented study. In this study, the authors combined phage panning and NSG sequencing to isolate monoclonal antibodies. Two minor questions are as follows: 1. what was your consideration when you selected these four cell lines S1T, HepG2, SKBR3, and KYSE520? 2. Why do you choose E. coli for VHH antibody expression that may cause false fold? 

Reviewer 2 Report

1. The part of Introduction devoted to the phage display description can be more concise. Instead, the authors should stress that they work with camelid antibody fragments, since it is not obvious for specialists not werll acquainted with the subject until they reache Materials and Methods section. 

2. Results section. 
Line 85: the dimensionality of the numbers 1.9x107 and 4.6x107 should be defined.Table 2. The terms "Parameters are not well understood.
Whether "Total number of read sequences" belong to VHH sequences or VHH sequences are termed as merged ones? What is the difference between "single occurrence" and "unique" sequences? What does "highest frequency" mean? 
Figure 3. The text in the rectangles is blind and not readable.
The legend to this figure (text in bold) is confusing. It seems that the phylogenetic tree concerns not cell line, but phage-displayed VHH fragments. The following legend text should be transferred to the body of the manuscript; the last  phrase of the legend is not well understood.
Figures 5 and 6 can be combined. Their legends are misleading as well: The presented results have been obtained by cell sorting and not by NGS.

3. Discussion
Lines  198-200 ("The enrichment of library phages ... non-target organic molecules [25]" - this phrase is unclear.
Line 234 and some others - "amplification fold factor" - does it mean amplification round number?
Line 241: the phrase "the cross-reactivity of the group G phage was confirmed by multiple binding" is not clear. What "multiple binding" is meant? 
The cross-reactivity and sequence similarity (and even identity) of phage-displayed VHH fragments obtained after immunizations by different cells and hence, prepared from different phage libraries are not enough discussed. However, it is a very interesting topic, which should become of attention for a further development of the methodology. The authors are encouraged to pay more attention to it rather than discuss amplification round numbers (amplification factors). 
The results of VHH production in E.coli are poorly presented, procedures for these experiments are poorly described, hence it should not be mentioned in this manuscript, leave it for the next one.

Materials and Methods. CDR3-specific primers should be presented, maybe in Supplementary materials.   

Data availability statement - the authors should define here, where NGS data for VHH sequencing are deposited.

Please check some phrase constructs and typing errors.
